# Gastronomy and Tourism: Socioeconomic and Territorial Implications in Santiago de Compostela-Galiza (NW Spain)

**DOI:** 10.3390/ijerph17176173

**Published:** 2020-08-25

**Authors:** Emilio V. Carral, Marisa del Río, Zósimo López

**Affiliations:** 1Functional Biology Department, Ecology Section, University of Santiago de Compostela, 27002 Lugo, Spain; 2Business Organization and Marketing Department, Marketing and Market Research Area, University of Santiago de Compostela, 15782 Santiago de Compostela, Spain; marisa.delrio@usc.gal; 3School of Education, International University of La Rioja, 26006 Logroño, Spain; zosimo.lopez@unir.net

**Keywords:** gastronomy, tourism, food demand, restaurant offering, normal food consumption, official gastronomic advertisement, foodshed implication, Santiago de Compostela, Galiza (Spain)

## Abstract

It is a worldwide well-known fact that gastronomic tourism does not always contribute to the cultural, social, economic, and territorial development of the host community. Therefore, its study requires a multidisciplinary and holistic approach to explore and interpret this phenomenon. From this perspective, the paper analyses the consumption of food products by tourists in Santiago de Compostela in Spain (2013–2014). Personal interviews (2081) with visitors and food industry establishment representatives were done. Compared with the normal food consumption of the Galician population, the food production capacity established by the corresponding Santiago foodshed calculation, and with the gastronomy official advertising (tourism web pages analysed by multimodal analysis), the gastronomic tourist experience is standardized and poor, limited to practically two products: rice with lobster, and octopus. This standardization supposes a high reduction in the diversity of the product that can be offered and produced in terms of proximity, and its territorial differentiation, comparing to the usual consumption of the Galician population, to the potential agricultural production by associated foodshed, and to the gastronomic advertising through official web pages. Thus, in this case, gastronomic tourism is not contributing to the social, economic, and territorial development of the host community or, ultimately, to the sustainability of tourism.

## 1. Introduction

Among the topics in tourism research, gastronomic tourism is one of the most important from different points of view and considering different meanings—Cultural, social, geographical, and political—so it is essential to adopt a holistic approach for its study. Food tourism, culinary tourism, and gastronomic tourism are often used as synonyms, but the relationships with cultural systems resulting from each of these concepts are different [1]. From this point of view, we understand that gastronomic tourism establishes a closer relationship with the concept of sustainability since it represents the role of food in the host culture; therefore, it allows us to establish, or at least investigate, the holistic relationship between tourism and the socio-community system of the host society: culinary (expression of cultural heritage and cultural consumption), forms of production, forms of social relationship, and local trade (economic, social, and environmental benefits).

Spain is the second most important tourist destination in the world in terms of number of visitors: 83.7 million foreign tourists travelled to Spain in 2019, according to [2]. The most recent study on their motivations [3] based on more than 55 million mentions of tourists on the Web-highlights their cultural attractions as the most important (28% of mentions on the Internet), followed by those that constitute the traditional model of sun and beach (19% of mentions). According to data from the same study, the “top 3” of tourist products is made up of “nature tourism” (11%), “active tourism” (10%) and “gastronomic tourism” (9%).

Diversity is the most outstanding feature of Spanish gastronomy. Garden products, oil, wine, or ham represent important items of Spanish exports, although they are always valued in international markets below their Italian or French competitors. The most typical dish—And a cliché—Of Spanish gastronomy around the world is the “paella”.

Galiza is located on the Atlantic coast of Spain (just above Portugal). Its climate is humid, and its attractions are far from those on which the sun and beach tourism model is based. Nature (the greenery) and gastronomy are the attributes most cited by its visitors [4]. Galician gastronomy is known for its wines, especially its white ones. It is also known for its variety of fish and seafood, and for the quality of its meats, converted into dishes by simple preparations.

Santiago de Compostela is located in the interior of Galiza and is its capital. It is a small (less than 100,000 inhabitants) medieval cities, declared a UNESCO World Heritage Site in 1985. Its international image is linked to the Camino de Santiago, a pilgrimage route of Christianity for twelve centuries. In this context, this paper seeks to assess the extent to which there is a relationship between food production and consumption at the local level. It also assesses the gastronomic supply and demand linked to those who visit Santiago de Compostela. The data obtained from the different sources used in this study provide indications of the sustainability of this city, that is, to what extent Santiago de Compostela has sustainable tourism.

According to the World Tourism Organization [5], “Sustainable tourism” is “Tourism that takes full account of current and future economic, social and environmental impacts to meet the needs of visitors, industry, the environment and host communities” and It proposes the following definition of Sustainable Gastronomic Tourism: “People who, during their trips and stays, carry out activities based on the tangible and intangible gastronomic cultural heritage in places different from their usual environment, for a period of time less than one year, with the main purpose of consuming and enjoying products, services, experiences and gastronomic inspirations in a priority and complementary way [6] (pp. 227–228). They help the development of the receiving society and to maintain in present and future time the preservation and safeguard of the Tangible, Natural, Intangible and Mixed Gastronomic Cultural Heritage, the endemic species, the environment and the food and economic security of a site, community, locality, region or country”. The same author states that in gastronomic tourism one cannot speak of sustainability if there is no system of gastronomic and tourist planning that prioritizes the benefits of local communities through a regional system that is born from the earth and ends up in it. By other hand, tourism and sustainable development is a very important and actual coupled issue, yet the United Nations General Assembly designated 2017 as the International Year for Sustainable Tourism for Development. The same year, UNWTO member states—At the 22nd UNWTO General Assembly—Signed the Chengdu Declaration on Tourism and the Sustainable Development Goals.

Therefore we can conclude, that we should start from the idea that tourism is a mass phenomenon with important social, cultural, and environmental consequences, and from this perspective, the study of tourism requires a multidisciplinary approach to explore and interpret this phenomenon rather than create it [7].

However, enhancing sustainability through gastronomic tourism would be complex, because sustainability itself is often conceived broadly within policy proclaiming the benefits of gastronomic tourism, with a need to study, characterized and understand the each dimension of sustainability: economic, environmental, social and cultural, which independently enhances sustainability.

Gastronomic tourism studies are mainly oriented towards case-studies in relationship with market segmentation, tourist behaviour, preferences and consumption, relevance of food/product image and the strategic role of gastronomy as a driver of tourist destination selection [8,9,10], or focused on research about the development of specific strategies to attract gastronomic tourists, as co-creation experiences [11], or more singular aspects, such as the role of restaurants in the development of gastronomic tourism [12], in relationship with specific food items: olives [13], home-made bread [14], cheese [15,16], wine [17], blue tuna [18] or ham [19]. A broad approximation could be the community-based tourism approach, but usually only in the sense of matching regarding the items offered as a part of community-based tourism [20]. The general perception is that the main research is narrowly focused on the topic of “tourists looking for an authenticity approach and sense of place” perspective [21].

A recent review [22] about food tourism studies in Southeast Asia, reflects, in a general way, the same facts enumerated above: the most applied approaches were focused on case/empirical studies, followed by comparative case studies, and only one paper was related with the analysis of government tourism web sites. Other studies present a broader perspective in relationship with the opportunity of gastronomic tourism for local development (as a local economic resource), but the study pays more attention to the linkage between destination image and food events [23].

In our case, research follows the idea of [24], understanding gastronomic tourism as a political capital, where food and tourism, with a historical connection, binds them together as a political force in a general way, which includes economic, social, cultural, environment and political aspects, without forgetting the evident and strong relationship between food and agriculture, and its related aspects with public policies and strategies. In the same sense as [24], we share the concept of food tourism as a visionary state, a vision of the future (or the present) in which gastronomy tourism offers a scalable cost-effective means of local and regional development, with the potential to enhance regional economy, local heritage and environment appreciation, as [5] reflects. This holistic perspective should be including a “meeting point” for researchers, stakeholders, tourists, the community and political organizations. In any way, this wide-range point of view is not new, yet [25] expressed that “the study of tourism phenomenon requires a multidisciplinary approach, it is hard to isolate it from other phenomena and we should try to explain and interpret it, and not create it”. In the same way, it talks about a “knowledge-based platform” as the starting point of projecting tourism from a perspective with more holistic, global, multidisciplinary, and trans-disciplinary vision [26]. The point of view expressed by some authors about the relationship between tourism, daily-life and food is similar, understanding this link as a tourism product and a tool for identity/brand place development [27]. In our case, daily-life food consumption analysis of Galiza adds a very relevant point, because it belongs to the Atlantic Diet concept: healthy, functional, and bioactive [28].

Following this holistic perspective, our research analysed the relationships, and their implications, between tourist food consumption, daily-life food consumption in Galiza, restaurant offerings and local/regional food potential production (foodshed analysis), in order to understand the general implications of gastronomic tourism facts in Santiago de Compostela. This wide-ranging perspective could help us to point out the several tasks that should be development by a holistic tourism policy, from the point of view of regional/country perspective, linked gastronomic tourism, food offers, agricultural production and community development.

Consequently, our study analysed the food consumption demand by tourists according to the nationalities with the highest frequency of visits, as well as the gastronomic offerings in the food industry establishments of Santiago de Compostela (Galicia-NW Spain). This consumption is compared with the normal food consumption of the Galician population and the production capacity of the corresponding foodshed to qualitatively determine the degree of sustainability of the gastronomic tourism activity: the cultural reflection of the local cuisine and the promotion of local production according to the gastronomic supply/demand relationship. This perspective allows us to assess which aspects are potentially related to the negative impacts of gastronomic tourism: homogenization-globalisation of the offer, loss of food heritage, little or no prominence of local food production, and concentration of economic benefits. On the other hand, these aspects are reinforced by the fact that local food consumption is an aspect that influences the tourist’s overall satisfaction with the trip [29]. The elements of cultural heritage and products from local agriculture [30] and of identity [31] are equally important for visitors. In addition, the promotion of these perspectives in gastronomic tourism management allows places to be more competitive globally [32]. Therefore, evaluating the sustainability of the gastronomic event/destination should not only be done from the strict economic point of view but should also be understood as an element of cultural production/consumption. The linkage of economic and cultural aspects also implies an environmental link from the perspective of local production, short distribution circuits, and associated sociocultural systems.

All this holistic process becomes more important when we refer to a city with a significant touristic component, as is the case of Santiago de Compostela, which is the end destination of pilgrimage routes and which has reached great international visibility as a tourist destination in the two decades of this century. This promotion, mainly channelled through official promotion policies and aided by the publication of different literary works and films, has resulted in the “reinvention” of the city as tourist attraction, with a translation into a significant increase in the number of tourist visits, going from less than 10,000 in 1990 to more than 250,000 in 2013 [33]. All these characteristics make the city the ideal setting to analyse the impact and behaviour of visitors in relation to the aforementioned factors.

## 2. Materials and Methods

The research objective, gastronomy in the city of Santiago de Compostela and its relationship with the different socioeconomic aspects mentioned above, is part of a broader project to analyse tourist uses and consumption and their influence on the local community. Specifically, in the present case, the study was approached from different angles that required the use of three complementary methods.

### 2.1. Technical Data

First, to understand the food supply in the city and its demand by visitors, a descriptive analysis of primary information on the characteristics of the suppliers and demanders of food provided by two samples was carried out (Table 1): one of merchants and another of visitors to the city.

As can be deduced from Table 2, the samples are balanced in terms of gender and average age, with people interviewed ranging between 16 and 91 years. Most of the respondents from both samples had a higher education. Professional activity on the part of merchants was linked to tourism (totally or partially). The merchants sample consisted of restaurants, bars, and food stores distributed throughout the city.

From the questionnaires characterizing the two sample populations, the variables shown in Table 3 were selected. Most of the items were formulated as open questions so that the biases of the research team would not influence—And, therefore, contaminate—The responses of those surveyed.

In the responses, all the information provided by the interviewees in their answers to each open question was taken into account. From the totality of the answers, different categories were extracted (identifying the basic concepts included in each of the answers and calculating the frequencies with which each concept was mentioned in the total sample). After following this procedure for all open questions, a variable was created for each of the identified categories. With the statistical package IBM SPSS vs. 24 (IBM, Armonk, NY, USA) [35], univariate and bivariate descriptive analyses of the variables under study were performed.

### 2.2. Data

Next, to understand the local customs on food, a descriptive analysis was carried out on the secondary information about the consumption of food products provided by the National Institute of Statistics of Spain through the Family Budget Survey regarding consumed amounts of food and beverages [36]. The data are separated by autonomous communities. Those relating to the autonomous community of Galicia, in which the city of Santiago de Compostela is located, were analysed. Likewise, and based on the data provided by the Galician Institute of Statistics (Survey of Crop Areas and Yields) and the Agricultural Statistics Yearbooks (Ministry of Rural Affairs), the productive potential for the Region of Santiago de Compostela was determined for the period studied [37].

### 2.3. Decoding Websites Data

To decode the information on gastronomy within the main online institutional sources of tourist information, the information published on three websites during the field study was analysed:Turgalicia.es (currently: turismo.gal) [38].Gastronomiadegalicia.com [39].Santiagotourism.com [40].

The first two sites depend directly on the tourism bureau of the regional government of Galicia. The latter belongs to the local government of Santiago de Compostela. In all cases, they are both governments’ means of institutional social communication on tourism.

For this work, it was necessary to analyse the meaning of the text, images, video, and hypertext reading of the contents of the aforementioned websites. For this, the use of a theoretical framework based on multimodal discourse analysis (MDA) [41] to achieve a content analysis of these websites that is as complete as possible. This multimodal approach is concerned with the meaning that is created through the different configurations, combinations, and interactions, in the same message, of image, sound, speech, text, typography, and/or melody ([42], p. 148). The MDA applied to websites proposed in this study is based on multimodal semiotic theory [43,44,45,46]. Authors such as [47,48,49] have already worked productively with an MDA model that was exclusively applied to website analysis.

As some authors notes, before describing and analysing the text, images, videos, colours, or animations of a tourist website, it is necessary to be clear about some concepts such as the “mode”, “semiotic source”, and “modal affordance” [50] (pp. 21–27). The denomination “multimodality” is related to the interrelation of the so-called modes; these are an organised set of semiotic sources for the creation of a meaning that is socially modelled and culturally transmitted ([51], p. 54). Modal affordance can be defined as what can be expressed and represented simply; that is, how a mode has been repeatedly used to represent something through social conventions in certain contexts and, in an unequivocal way, refers to the same meanings. For example, one would have to think about how gastronomic products are represented on a tourist website: one possible mode is the display of fresh produce in which there is no room for misunderstanding; another mode is the sample of the processed product in the form of a cooked dish as a final product, which we usually find on a menu or advertising device and that the consumer expects to find. Both the menu of a restaurant and its website or its advertising spot—if it has one—are units composed of numerous modes and semiotic sources. We can generally call these units composed of numerous modes and semiotic sources “multimodal artefacts” [52]. Taking the latter into account, we can affirm that tourism websites are multimodal and multimedia artefacts created in a specific sociohistorical context within a “semiotic landscape” ([53], p. 35) that tend to compile different semiotic sources, such as videos, images, logos, text, audios, and digital publications, such as brochures, dossiers, reports, and catalogues. Like all multimodal artefacts, an official tourism website is created in a context of specific interests and with specific purposes ([54], p. 136).

In turn, it can be asserted that there is no perfect way to recover and preserve an object of study itself given the digital nature of the product ([48], p. 172) since the environment, context, and hyperlinks of a website can break, be moved, or simply disappear, which means that its design and functionality have been changed. Being aware of the constraints in this field and when retrieving digital files located on the World Wide Web, the Wayback Machine tool from archive.org was chosen to retrieve the information from these sites between March 2013 and March 2014 [55].

## 3. Results

Owing to the coexistence of different methods to approach the relationship between tourism and gastronomy in the city, the main results are presented in sections, reserving their joint interpretation for the Discussion.

### 3.1. Demand, Supply, Consumption, and Production of Food in Santiago de Compostela

#### 3.1.1. Food Purchase and Consumption by Visitors

The importance of gastronomy for people who visit Santiago was demonstrated in their answers to the first question that was asked: A majority (51%) of those who answered this question indicated that they planned to eat or drink something during their stay in Santiago. This figure also represents 31.85% of the total number of visitors surveyed. Regarding the preferred type of food, a strong preference for shellfish and octopus was observed (Table 4). Interestingly, Santiago de Compostela is a city located inland. Among the respondents from the countries with the most visitors during the study period, Spaniards were those who most frequently opted for the consumption of octopus and shellfish over other types of food (Table 5), and in general, the preference for these products decreased as the visitor travelled more times to Santiago, with their food type preference becoming more similar to what is typical for the Galician population (Table 6).

Regarding the places in which visitors declared having eaten, it was observed that, as with food, the demand in this case was concentrated in very specific spaces (Table 7). Almost 70% of the answers referred to a restaurant in the old town area. A single restaurant, “Casa Manolo” (which stands out for its low prices), was mentioned by tourists 134 times.

To make their decisions about what and where to eat in Santiago, only 11.00% of visitors consulted some source of information. If the percentage is limited to visitors who answered this question, it goes up to 19.5%. Word of mouth made up 79.35% of the sources of information consulted (Table 8). Of these, 23.85% refer to suggestions from contact personnel in the tourism sector (guides, receptionists, and tourist information offices). Only 10.09% declared using the Internet as a source of information (exclusively or in addition to other sources).

Food purchases were also concentrated around a few products and areas. A single product, the Santiago almond cake appeared in approximately 30% of the responses. The top three products made up almost half of all mentions (Table 9).

The purchase of food products, as well as their consumption, by visitors was conditioned by the place of origin of the interviewees and by the frequency with which they visited the city. If we look at the place of origin, the pattern of behaviour of Galicians differed from the rest: They bought Santiago almond cake, cheeses, and wines with significantly less frequency in favour of other sweets and bread (Table 10).

Looking at the frequency with which the tourists had visited the city (Table 11), the observed food consumption pattern was repeated: Frequent visitors bought the top-ranked products—Santiago almond cake and cheeses—significantly less often and bought other sweets and bread significantly more. In this case, no significant differences were observed in the purchase of wines.

The demand for these products (Table 12) occurred mainly in different enclaves of the Old Town (66.81% of the responses alluded to them). The most mentioned retail format was the supermarket, followed closely by the tourist shops of the Old Town.

Only 3.3% of the visitors sought information about the commercial offer of the city before making their purchases. The figure rises to 6.3% if it is calculated based on the number of people who answered this question. In relation to the sources consulted, only 54 people responded (Table 13). As in the case of the food industry offer, the majority (65.67%) referred to word-of-mouth information, with 24.07% mentioning contact personnel in the tourism sector and 12.96% mentioning the Internet among the sources of information consulted.

When visitors were asked to add opinions about the city’s gastronomic offer to their responses (Table 14), the theme alluded to most was its variety, although opinions were divided between those who perceived it to be varied and those who considered the offer to be too homogeneous. In addition to the latter, there were those who thought that the gastronomic offer catered too much to tourism. The opinions of visitors on this issue are therefore conflicting.

The next-most mentioned aspect of the service was the price. Again, opinions were divided, between those who considered them appropriate and those who did not, although most tended to find them appropriate. Something similar occurred with the sufficient/insufficient supply dichotomy, although in this case, the balance was even more towards the positive. The quality of the products was one of the attributes most often deemed positive and least frequently deemed negative. Finally, the responses that addressed the gastronomic offer in general were positive in almost all cases.

#### 3.1.2. Supply of Food Products

From the perspective of supply, a process of “gastronomisation” (concentration and spatial dominance of specific food industry businesses) was observed: bars, restaurants, and food shops were concentrated in the Old Town and the New District (Ensanche) (Table 15).

In the Old Town of the city, 60% of the businesses in the sample stated that they mostly or exclusively served visitors. Despite the typology of the clients, 74.6% of those consulted did not find differences between visitor demand and the preferences of the locals. Perhaps because of this, the merchants almost never changed their offer during the tourist season (94.4%). A total of 32.4% of respondents indicated that they would prefer another type of tourist: essentially, a tourist with more purchasing power.

The interviewees were also asked about their best-selling products. They were allowed to identify up to six categories of “top products” (Table 16). The responses indicating “drinks” or “infusions” exceeded 37% of the total. If generic responses such as “food” or “prepared meals and menus” were excluded, the following stood out in order of importance: “meats and charcuterie”, “fruits, vegetables and legumes”, “pizza and pasta”, “octopus and shellfish”, “cakes and sweets”, “bread”, and “fish”.

#### 3.1.3. Food Consumption in Galician Households

The findings for visitor demand contrasted with those obtained from the search and processing of secondary information on food consumption in Galician households (Table 17). Fruit, milk, vegetables, meat, and bread were the preferred foods consumed in Galicia. On the other hand, shellfish and octopus were foods of lower and markedly seasonal consumption, both individually and with respect to total food consumption (Table 18).

#### 3.1.4. Production Potential for the Region of Santiago

From the point of view of the productive capacity of the Santiago de Compostela foodshed (the territory defined by a radius of 50 km from the city of Santiago), we can conclude that there was sufficient agricultural production and surplus production of fruits, potatoes, vegetables, legumes, and meat, with a surplus of 100,392 ha for a territory of 369,780 ha, and a population of 373,976 inhabitants. However, there is a marked deficit about 3166 ha in the production of cereal grains [37].

### 3.2. Analysis of Institutional Gastronomy Websites

As mentioned in the methodology section, to analyse the information on gastronomy on the Web, the following official websites were chosen: Turgalicia.es (currently: turismo.gal); Gastronomiadegalicia.com and Santiagotourism.com.

#### 3.2.1. Turgalicia

The website was analysed between March 2013 and 2014, a period in which its content was dictated by the advertising strategy set by the tourism department of the Galician government, called Turgalicia, in the studied chronological period. The campaign designed by the advertising agency CIAC, which was called “Galicia, will you keep my secret?”, was presented in December 2010 [57]. According to the news taken from the specialised blog casosdemarketing.com, this campaign consisted of a television spot (previously teaser) in 45″, 30″, 20″, and 10″ versions. In the video, an intimate woman’s voice speaks on behalf of Galicia and confesses the “sins” that Galicia offers to its visitors. Along with the spot, the campaign had several originals for the press that sought to provide an image of renewed visual identity, but using as a starting point the Galician cultural roots, such as the Gaelic and Celtic codices, including radio spots and an Internet campaign within the Turgalicia website (www.turgalicia.es). The Turgalicia website (Figure 1) is therefore part of this advertising campaign that began in 2011 and lasted until 2014.

The website showed two options related to gastronomy:A hyperlink called “Festivals of interest” accompanied by text and an image of a square plate with what appeared to be cheese cut into portions and a loaf of bread (Figure 2). The text that accompanied this image was as follows:“Festivals of interest. Nearly a hundred Galician festivals have been recognized as being of interest for international, national, or Galician tourists. A world of festivals of all kinds, from gastronomic to folkloric as well as religious or the “rapa das bestas” (“capture of the beasts”). Which ones are celebrated this month? Choose one and experience it!.”

Figure 2, of a neat and orderly arrangement of elements on a plate that seems to be made of ceramic, has nothing to do with the plates served in the more than 3000 popular festivals that were celebrated in Galicia in a year ([58], p. 42), which could lead to false expectations by the consumer that resulted in dissatisfaction if they did go to one of those festivals.

A hyperlink with the name “Where to eat” within the resource locator of the site itself (Figure 3). The options focused on gastronomic offerings of fixed and regulated establishments such as restaurants. The text and options for choosing a restaurant on this website were as follows:

“In Galicia, there are some things that you can miss, but you cannot miss its gastronomy. Try the seafood: barnacles, king crabs, velvet crabs, lobsters... The clams and mussels are delicious. And the fish stews are unbeatable. Inland, the octopus, the empanadas (pies), the stews, and the pork shoulder with turnip greens are a must. Choose the restaurant below or use the search engine on the side for more specific results.”

The information shown in both links may lead one to think that in Galicia there are no other alternatives—Such as gastronomic festivals, casual venues that are popularly known as “chiringuitos”—Or other popular fairs in which food is traditionally served.

Additionally, this website allowed the visitor to download several tourist publications catalogued by more specific topics. Related to gastronomy, there were five publications: One on the Camino de Santiago, one on popular festivals, one on cheese and wine, and two specifically on wine, thus highlighting the importance given to wine tourism compared to other varieties gastronomy of the autonomous community. We must highlight a section from the publication called “Galicia. The Way of St. James”; on page 11, there was text that referred to the gastronomic offer apparently typical of the city of Santiago de Compostela:

“Tapas dining along Rúa do Franco (Franco Street) and A Raíña (Raíña Street). As this is your first night in Santiago, we suggest that you opt for the typical Galician portions for dinner. This is a good way to enjoy some all the essential dishes of Galician cuisine, such as octopus cooked feira style, stewed meat ao caldeiro, Galician pie (empanada), marinated pork loin (raxo), Galician chorizo-spiced pork loin (zorza), and pig’s ear; shellfish from the Galician estuaries such as mussels, cockles, and clams; or seasonal products such as Padrón peppers or xoubas (small sardines). You can try all these delicacies in the many taverns and restaurants of the old quarter. Rúa do Franco and A Raíña house the majority of these establishments.”

What may seem striking in this first text is the lack of beef options, limited to “stewed meat ao caldeiro”, the complete absence of goat products, or that pork products are represented by the “zorza”, the “raxo” and the pig’s ear, which are made to be consumed as a portion or tapa. It leaves out such complete, traditional dishes as the Galician stew or pork shoulder with turnip greens that incorporate local vegetable products, not to mention the most avant-garde alternatives based on market products. In summary, the content clearly invites visitors to the city of Santiago de Compostela to take a tapas tour on Rúa del Franco and A Raíña (Table 7).

#### 3.2.2. Gastronomiadegalicia.com

Within the Turgalicia website, there is a direct link to a page called gastronomiadegalicia.com (http://www.gastronomiadegalicia.com) (Figure 4) This autonomous website is part of the institutional communication of the body responsible for tourism promotion of the Galician government. Its content is exclusively geared towards Galician gastronomy. The content of the information varies, being adapted to the different seasons, and the food seems to change with each season, which would favour sustainability through the balance between the local production and consumption of food.

As seen in the menu on the right on the second page of this website (Figure 4), the options offered for browsing were the following:

Guide to Galician Food Products/High-Quality Galician Products/Gastronomy of the Sea/Typical Galician dishes/Gastronomic festivals of interest/Hotel Management School of Galicia/Publications/Other websites of interest/Protagonistas Newspaper Archive/Conocer y Harmonizar Newspaper Archive.

In each of the tabs, reference was made to products produced in Galicia, with a hyperlink dedicated exclusively to “Gastronomy of the Sea”. It was the only link dedicated to a type of product. In turn, in the navigation options, the search was done by product type and not by geographical area or season.

In the tab “Guide to Galician food products”, there was an abundance of images of fresh, live, or unprocessed products (Figure 5). This was a trend of the site: Of the 68 images that appeared throughout the different pages that made up the site, only 17 were prepared dishes. The processed included wine, bread, honey, and cheese (six images), but the images corresponding to fresh or live products totalled 31. Obviously, in the Typical Galician Dishes tab, all the images included in this document were prepared dishes given that it was a space intended for the explanation of cooking recipes.

#### 3.2.3. Santiagotourism.com

On the date of archiving (25 March 2013), the page provided information about the Catholic holiday of Lent and the Holy Week. According to the Catechism of the Catholic Church, people practising this Christian rite perform penance for forty days (from Ash Wednesday to the eve of Easter Sunday) in commemoration of the forty days that Jesus wandered through the desert after being baptised. This penance is materialised by inviting fasting and avoiding meat, especially on Fridays. This religious context explains why the gastronomic offerings announced on the site at this time of year are, above all, fish, and sweets.

One of the visual elements related to gastronomy was the composition “Santiago Pasión”, in which a plate of cooked fish could be seen in the background (Figure 6). The very name “Santiago Pasión”—In conjunction with cooked fish—Refers to the period of the Holy Week. It plays with the polysemy of the word “passion”, which can mean both appetite for something and devotion to something, like the torments suffered by Jesus Christ from the Last Supper to his Crucifixion

Within the home page of the site, there was a fixed hyperlink that was maintained in the base structure and referred to “Gastronomy” accompanied by an image of a woman who sells cheese (Figure 7).

Inside, the gastronomy website of this site had the following hyperlinks (Figure 8): Activities and gastronomic events/Food Market/The menu of Compostela/Where to eat/Taste to take home/Training/Gastronomic route.

In June 2013, another hyperlink, “Traditional markets near Santiago”, was added; it referred to all the traditional markets held in towns less than 30 kilometres from the city of Santiago de Compostela. Although the heading of this section read *“[…] a cuisine admired for the superb quality of its ingredients, of the sea and of the land, in which there is room for the most traditional to the most innovative […]”.* What was shown in the images were processed or cooked foods and where they could be eaten or bought. Information on the commercial activity of buying/selling food products was given priority over the culture and productive identity of the area.

The food market of the city had great prominence in this tab, as (1) there was a specific hyperlink dedicated to this traditional market, and (2) throughout all the other places within the site, this market was alluded to (such as in the hyperlink “Taste to take home”).

Another element that drew attention was the hyperlink on “Training” in gastronomy, through which the visitor could access data on regulated training in hospitality (both secondary and higher education) or on the Association of Hospitality Businesses of Santiago de Compostela and Region. In this part, the activities offered by other organisations (public and private) related to training in gastronomy and not in hospitality were ignored.

Within the website “The Menu of Compostela”, three options are presented in the form of a hyperlink: From the sea to the land/The kingdom of shellfish/Desserts, wines, and spirits.

The value of the meat, vegetable, and dairy products of Santiago and the region was omitted. It is worth mentioning that after clicking on the hyperlink “From the sea to the land”, veal and pork were mentioned by way of the traditional dish known as Galician stew.

## 4. Discussion

The analysis of the model of consumption of food products by visitors to Santiago de Compostela indicates, first, that gastronomic activity plays a fundamental role in the demands of tourists, given that it was mentioned by 32% of the people surveyed as a single factor and by 51% of those who expressed their activity preferences for their visit to the city.

The visitors show a special preference for the consumption of two products, shellfish and octopus, and for the purchase of three others, Santiago almond cake, cheese, and wine. Among the countries with the highest influx of visitors in the period studied (Brazil, Portugal, Galicia, and Spain), it is the Spaniards that clearly show a greater affection for the foods. It is also important to note that as the number of times an individual has visited the city increases, gastronomic preferences increasingly shift from the top products mentioned toward the normal consumption of the residents of Santiago and Galicia (meat, fish, bread). This behaviour may be due to a process of familiarisation with the aforementioned products [59]. In addition, this change in behaviour favours the sustainability of tourism and gastronomy, since the analysis of the Galician foodshed demonstrates the ability to produce (with surplus) the food products consumed regularly by the resident population.

In terms of the spaces, the “gastronomisation” of the Old Town of the city is observed: most of the bars, restaurants, and food stores that acknowledge—approximately 60% of them—selling exclusively or mostly to visitors are located in this area. They do not identify differences between the preferences of visitors and those of the local population, so they do not vary what is on offer in the high season. Thus, what is generally on offer at these establishments responds, throughout the year, to the gastronomic varieties that both customer segments demand.

On the other hand, the study of the three sources of tourist information on Galicia, and the city of Santiago in particular, indicates a clear bias towards the consumption of shellfish and octopus, although not exclusively, since in some cases cheese, bread, meat, and fish are also mentioned, but mainly as very specific products of specific occasions (gastronomic festivals). In the specific case of information to pilgrims, the institutional website of the Autonomous Government recommends in its brochure “Galicia. Camino de Santiago” that the visitor tries dishes that are usually served in the form of tapas in the establishments located in the streets of the city most overcrowded by visitors.

The impact on the general behaviour of online content visitors can be considered relatively low, since less than 20% state that they have used the Internet to learn about the city’s food industry offerings and approximately 13% have used it to learn about their commercial offerings. Thus, an appropriate orientation of institutional advertising would lead to a much more sustainable model, not only from the environmental point of view (local production and consumption) but also in terms of the promotion and recognition of local agricultural communities and their role in environmental care, the production of healthy food, and the generation of culture associated with the use of the area and its natural resources.

These results indicates a clearly reductionism landscape for the gastronomic tourism fact, because following [60] Gastronomic tourism can be describe as the dynamic process of “Visiting primary and secondary food producers, gastronomic festivals, restaurants and specific places where tasting dishes and/or experiencing the attributes of a region specialized in food production is the main reason for a trip”. After that, the case of Santiago de Compostela laid so far from the real, and desirable, concept of gastronomic tourism, and we could talk about the Santiago gastronomic tourism fact as “bulk food tourism”, a merely food offering and consumption which not reflects, in any circumstances, the community gastronomic daily-life, neither the agri-food production potential of the country.

## 5. Conclusions

There is a clear trend towards consumption and practices related to gastronomic activity among visitors to the city of Santiago de Compostela during the time analysed. This activity is specially focused on the consumption of shellfish and octopus and on the purchase of Santiago almond cake, cheese, and wine. These products differ widely from those normally consumed by residents in the city and in Galicia. In comparison with normal consumption (vegetables, meat, fish, bread) and the production capacity of the foodshed of the region of Santiago, the gastronomic preferences of visitors do not correspond to a model of tourism sustainability, clearly being counter to the promotion of local production and consumption. On the other hand, the institutional information supplied to potential visitors in terms of gastronomy only partly reflects the varied offer of existing food products, giving seafood and octopus products outsized prominence. This information clearly stresses (product photographs) a cognitive approach—more than an affective one—when the affective contents would be more appropriate to shape a powerful, clear, and long-term image of the tourist destination [61,62]. Therefore, these tools should be managed to promote a more varied gastronomic offer, which truly represents the normal consumption of food products by the resident population and that also favours the local production capacity of these food products within the corresponding foodshed. Another important tool, in view of the results, could be institutional promotion of a gastronomic offer directed by the restaurateurs, given that, as we have seen, their suggestions are acted on by a significant percentage (23.85%) of the visitors, who refer to their sources of information—About what and where to eat—As word of mouth.

The more times a visitor comes to the city, the closer his gastronomic demand gets to the foods normally consumed by Galicians. This trend indicates that an adequate reorientation of sociocultural planning, specifically gastronomic planning, would lead us to a greater sustainability of the system: greater local production/consumption and better knowledge of the socioeconomic reality of the host community.

From the methodological point of view, the research shows the convenience of holistic approximation applied to the gastronomic tourism fact analysis, because this perspective let us point out several disfunctions or gaps in the relationship between visitors food consumption, food offering and governmental gastronomic marketing, and of course, with all others items in relationship with the local community development, local agri-food production, and its economic and environmental related aspects (sustainable development).

Even more, some basic and useful aspects used for this approximation, as foodshed study or day-life food consumption characterization, doesn’t need field-experimental work (expensive), because to get a first view, it is enough work with statistical information, provided and open accessed, by the governmental institutions.

The results of this work may be representative of the situation in many tourist destinations that lack a sustainability objective in terms of gastronomy. The results of the extensive research carried out by [63] in the Canary Islands serve as an example, from which it is concluded that, according to the comments made by visitors to TripAdvisor, the two restaurants of the Hard Rock chain are the most highly valued of the islands’ gastronomic offer. So, is there a gastronomic offer? And if there is and it is, does it serve to give a differential value to the Canaries, or does it use the territory’s resources, where are the suppliers, where does the food travel from, how many indirect jobs does it generate, in short, is it a sustainable offer? Finally, why do so many tourist destinations “suffer” from a globalized gastronomic offer? The same questions can be asked for Santiago de Compostela.

The reason seems to lie in the consideration of tourism management as a zero-sum game, in which what is gained in sustainability is lost in profitability. This belief is based on the idea that tourists in the destination demand “what they want” and public and private managers must limit themselves to satisfying them. The results of this study indicate that tourists demand “what they are offered”. Demand adapts to what is offered, and supply believes it adjusts to what satisfies demand, without prior knowledge of what tourists’ value or would value if it were made known to them.

The results obtained in previous studies contradict this perspective. Tourists are more satisfied when they consider that they contribute to the sustainability of the destination to which they travel [64,65]. Moreover, recent studies support the existence of a positive relationship between perceived authenticity in a destination and tourist satisfaction [66]. A gastronomic offer that is consistent with the reality of the destination must necessarily be as authentic as possible, if it is well communicated.

Tourism is, in general, a leisure activity. Most tourists do not know the destination in depth before the trip (image—Not even after the trip (“real” image). It is the destination—Its managers—Who must take an active role in delivering a message based on what the destination is, rather than what they “think” it is. The results of this study show that the institutional communication of the destination analysed tends to reinforce the “clichés” about fate. The theory of cognitive dissonance [67] would advise avoiding messages that contradict the initial image; however, there is room for messages that—without contradicting it—complete it with a discourse that is more adapted to what the destination is, and what it needs from tourists in order to continue to exist. This type of message would improve the authenticity perceived in the destination—and, as a consequence, the satisfaction of the tourists—would respond to the tourists’ oblative motivations, would increase their satisfaction and would change the “real” image of the tourists after the visit (achieving a “real” image, more similar to reality).

As for the approach to reality, the results of this study are based on objective data obtained from national and regional statistics. They show that food consumption in the city is “sustainable” in the sense of [6]: most food constitutes a cycle, that is born on earth and dies on it. The use of objective sources of information is essential to draw the reality of the destination if we consider that, from a subjective point of view, it is difficult—or impossible—to describe the reality.

The implications of the results of this article are not positive if evaluated in relation to their contribution to the five objectives, that the [5] proposes for gastronomic tourism in its agenda 2030: (1) Economic growth could continue in the medium term, but without the adjectives “inclusive” and “sustainable”, insofar as the results of the activity benefit mainly a few; (2) the positive impact on employment and poverty reduction would be in the hotel and catering industry and in trade (seasonally), but would not impact sufficiently on employment in the primary sectors; therefore, inclusion would not be favoured; (3) the efficient use of natural resources would not be taking place, which would have negative effects on the environment—the abandonment of areas for cultivation and livestock breeding with the consequent increase in forest mass and fires—and would accelerate climate change; (4) the cultural values, diversity and intangible heritage implicit in Galiza’s gastronomic wealth would not be preserved; and finally (5) mutual understanding would not be promoted because interaction with visitors would take place at a very superficial and standardised level.

From the management point of view, the results of this work should be useful for managers and businessmen in the analysed destination, which could be extended to public and private decision-makers in other destinations.

## Figures and Tables

**Figure 1 ijerph-17-06173-f001:**
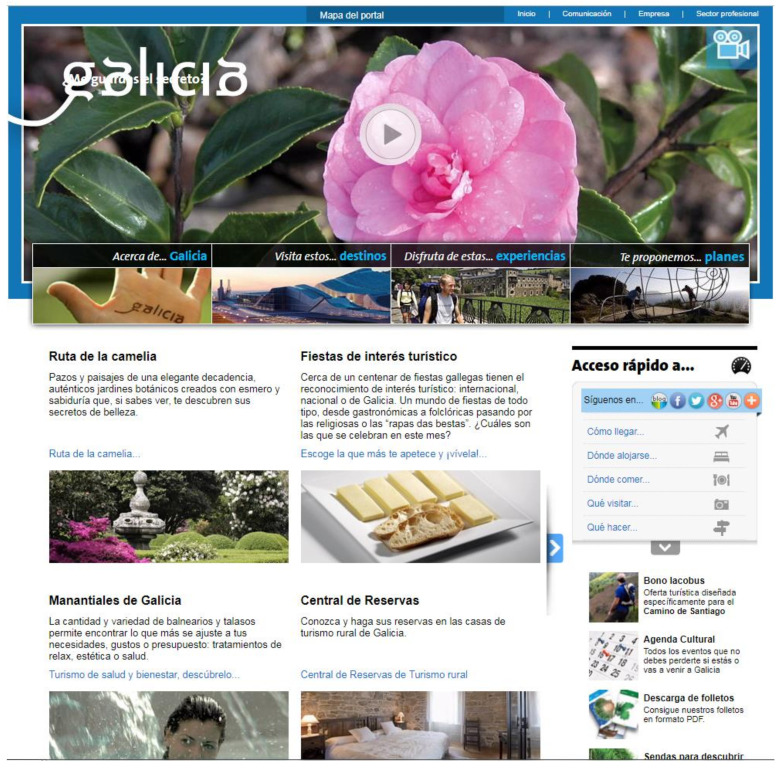
Images from the landing page of the website www.turgalicia.es. Source: [38].

**Figure 2 ijerph-17-06173-f002:**
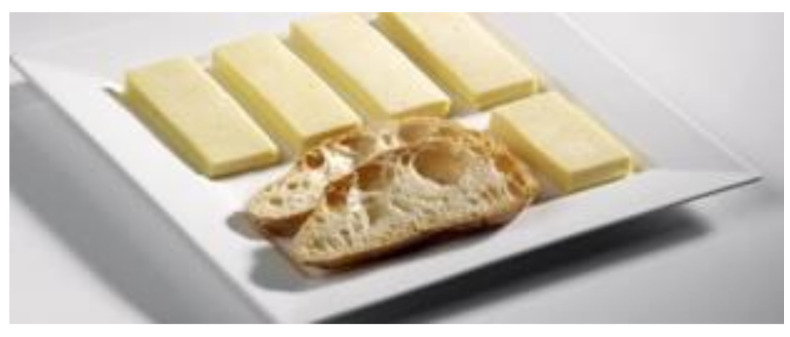
Image that accompanies the hyperlink “Festivals of interest”. Source: [38].

**Figure 3 ijerph-17-06173-f003:**
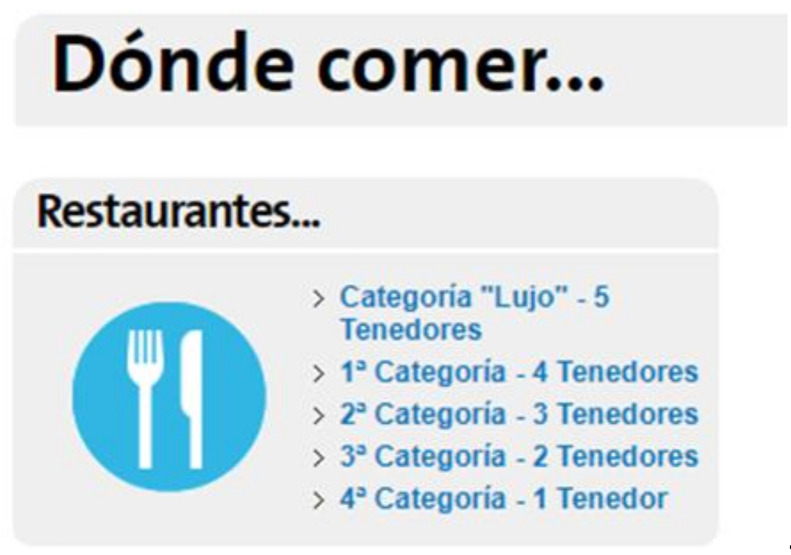
Image that accompanies the hyperlink that shows available restaurant information. Where to eat? Source: [38].

**Figure 4 ijerph-17-06173-f004:**
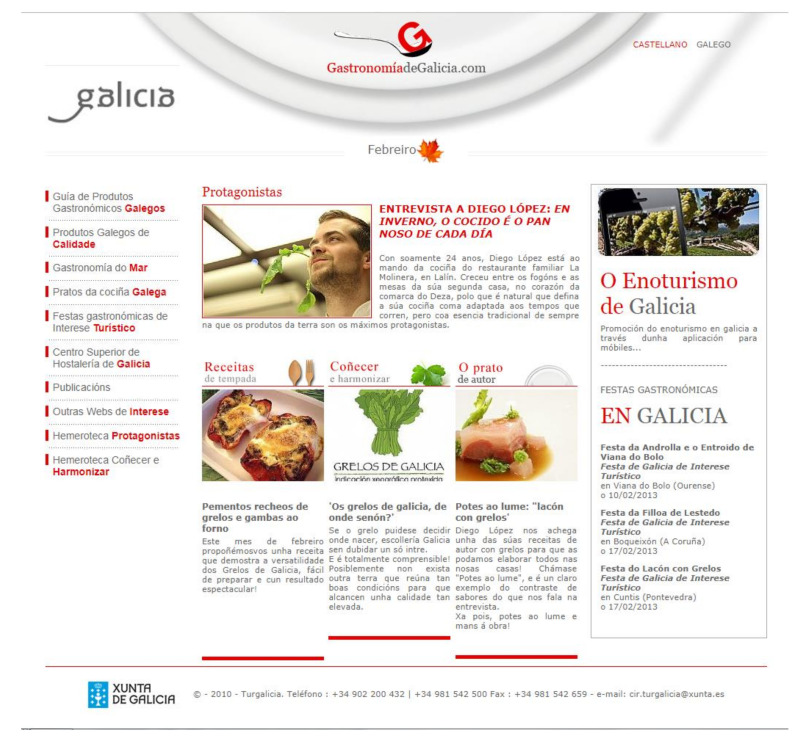
Landing pages of gastronomíadegalicia.com. Source: [39].

**Figure 5 ijerph-17-06173-f005:**
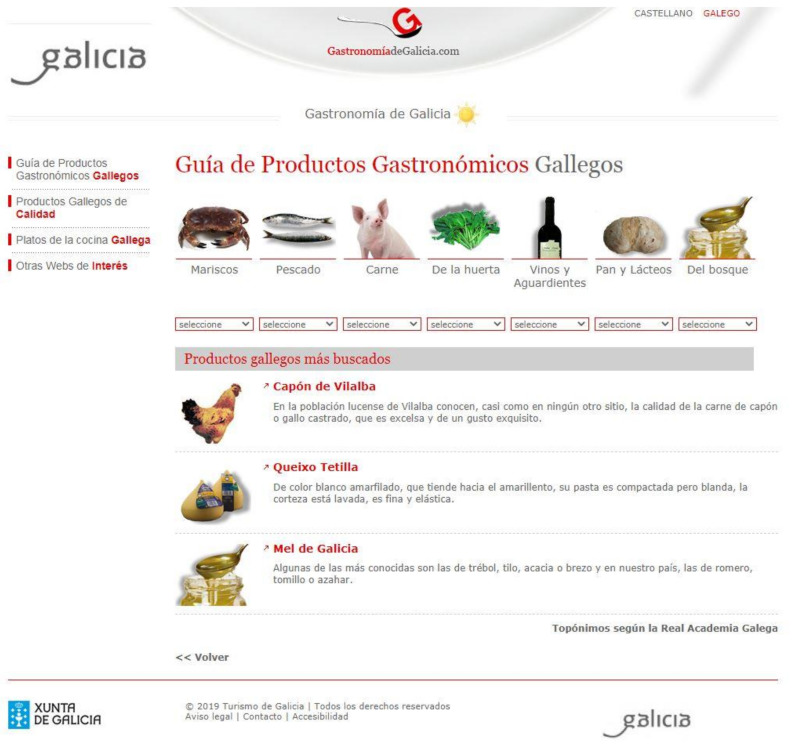
Guide to Galician food products within www.gastronomiadegalicia.com. Source: [39].

**Figure 6 ijerph-17-06173-f006:**
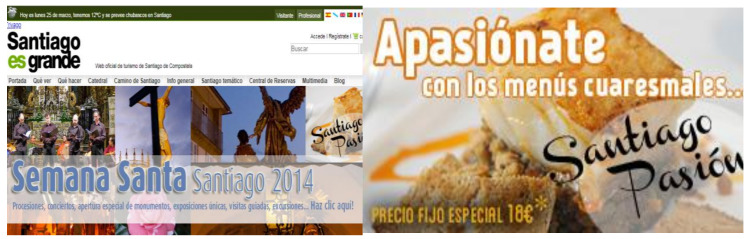
Detail of the header of the Santiagotourism.com site. Source: [40].

**Figure 7 ijerph-17-06173-f007:**
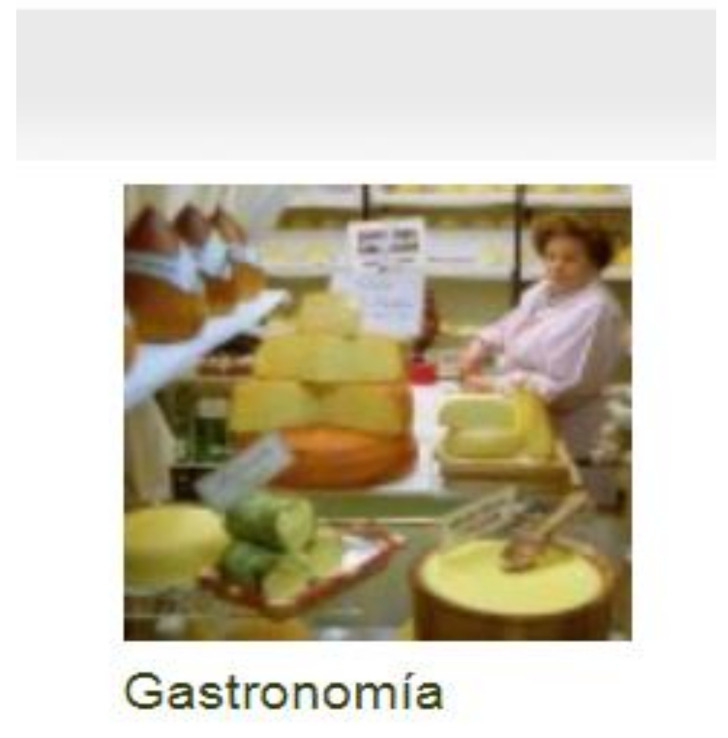
Detail of the hyperlink and image that alluded to the topic “Gastronomy”. Source: [40].

**Figure 8 ijerph-17-06173-f008:**
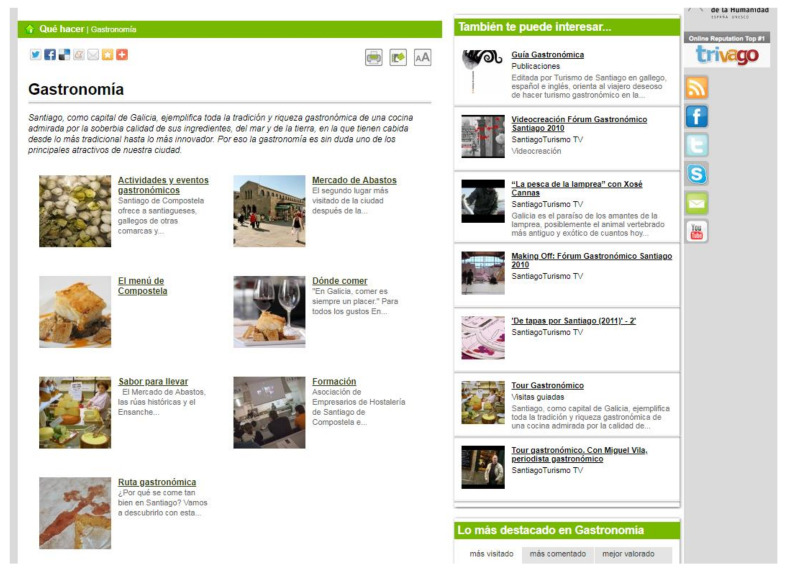
Detail of the menu within the “Gastronomy” website. Source: [40].

**Table 1 ijerph-17-06173-t001:** Technical Data of the Analysed Samples of Visitors and Merchants.

Survey Technical Attributes	Visitors Data Base	Commercial Sector Shops
Survey type	Personal (structured questions)	
Universe	Visitors to Santiago: >16 years old, from Galiza, Spain, Brazil and Portugal. ^1^	Restaurant, bar and food shops in Santiago de Compostela
Statistical population	Infinity (>100.000)	
Sample size	2081 valid surveys (Galiza: 398/Spain: 878Portugal: 408/Brazil: 396).	1362 (reference sample)
Sampling error	Whole sample: ± 2.15%For different countries: Galiza: ± 4.91%/Spain: ± 3.31% Portugal: ± 4.85%/Brazil: ± 4.92%	118 valid surveys
Confidence level	%; p = q = 0.5	± 8.63%
Survey development	On convenience by country origin quotas	
Dates	27 March 2013/26 March 2014.	On convenience by city districts ^2^

^1^ During the one-year period studied, Portugal and Brazil are the countries with a greater number of visitors, one from Europe and the other from the rest of the world. Data from Centre for Tourism Studies and Research-University of Santiago de Compostela [34]. 2 Districts of Santiago de Compostela: Old Town (Historic Centre); “The New District” (developed during the 1950-1960-1970 decades); Peripheric Neighbors; Industrial and Commercial Estate; and others traditional/historic settlements joined to the urban net in the last decades. The pilgrim’s entrance route to the city were differentiated too: entrances for the French (East direction) and Portuguese (South direction) pilgrim routes, and the pilgrim way out to Finisterre (West direction, to complete the pilgrimage route).

**Table 2 ijerph-17-06173-t002:** Socioeconomic Characteristics of the Interviewed Population.

Characteristics		Visitors	Commercial Owners
Gender	Female	57.8%	50.8%
	Male	42.2%	49.2%
Age		Mean: 41-year-old (SD: 14)	Mean: 40- year-old (SD: 11)
Educational background	University	26.2%	26.2%
	Higher technical school	5.5%	12.7%
Pilgrim	Yes	28.4%	-
1st time in the city	Yes	31.2%	-
Commercial activity by city district	Old Town	-	24.6%
	New District	-	16.1%
	Industrial estates	-	10.2%
	Santiago Pilgrim way	-	27.9%
	Countryside		3.4%
	Remain city	-	17.8%

SD: Standard deviation.

**Table 3 ijerph-17-06173-t003:** Study Variables of Visitors and Merchants.

Visitors	Merchants
Were you planning eating/drinking something in Santiago? ^1^	Yours customers are mainly composed by: Local people (not frequently) Mainly visitors Mixture of local people and visitors Mainly local people Visitors (not frequently) ^3^
Which did you eat/drink in Santiago? ^2^	Would you prefer other type of tourist? ^1^
Where did you eat? ^2^	Which are the bestselling products? ^2^
Which was your source of information about eating/drinking? ^1^	Are demands from local people and visitors different? ^1^
Did you purchase some food product in Santiago? ^1^	During the touristic high season, do you change the supplies offering? ^1^
What kind of food? ^2^	If yes, which were they? ^2^
Where did you purchase the food products? ^2^	
Which was your information source about food-supply facilities? ^1^	
Other opinion about food-supply offerings? ^2^	

^1^ Dichotomic closed variable, ^2^ Open and nominal variable, ^3^ Nominal, closed and polychotomy variable.

**Table 4 ijerph-17-06173-t004:** Preferences of Food Consumption by Visitors.

Food Choice (Answers: 733)	*n*	%
Shellfish	186	25.38
Octopus	165	22.92
Meat/Fish	112	15.28
Typical food	114	15.56

Number of answers (*n*) and percentage (%).

**Table 5 ijerph-17-06173-t005:** Preference in the Consumption of Food Products by Country of Origin.

Food Choice (%) *n* = 115	Galiza %	Portugal %	Brazil %	Spain %
Shellfish	1.00	2.94	9.34	15.14
Octopus	2.76	4.66	4.54	13.6
Other food	12.37	11.76	9.30	6.83

Number of answers (*n*) and percentage (%). Differences at level *p* = 0.05.

**Table 6 ijerph-17-06173-t006:** Preference in the Consumption of Food Products as a Function of the Number of Times Visiting the City.

Food choice	Less One Time/Year *n* = 1478	One Time/Year or More *n* = 478
Shellfish	11.30	3.14
Octopus	10.01	3.56

Number of answers (*n*) and percentage (%). Differences at level *p* = 0.05.

**Table 7 ijerph-17-06173-t007:** Places Chosen by Visitors to Eat.

Location	*n*	%
Restaurant- Old Town	446	69.15
Hotel/Hostel	62	9.61
Family/friends home	54	8.37
Urban green spaces	19	2.95
Bus/train station, airport	19	2.95
Hamburgers	16	2.48
Shopping mall	12	1.86
Restaurants- New Quarter	10	1.55
University- Hospital	7	1.08
Total answers	645	100
Don’t know/Don’t answer	1436	
TOTAL	2081	

Number of answers (*n*) and percentage (%) in relationship with Total answers.

**Table 8 ijerph-17-06173-t008:** Sources of Information on Food Options Consulted by Visitors.

Information Sources	*n*	%
Friends, relatives, fellows	91	41.74
People by the street	30	13.76
Official group guide	21	9.63
Hotel staff	20	9.17
Pilgrim/Tourist Office	11	5.05
Street banners	9	4.13
Internet	22	10.09
Other sources	18	8.26
Total answers	218	100
Don’t know/Don’t answer	1864	

Number of answers (*n*) and percentage (%) in relationship with Total answers.

**Table 9 ijerph-17-06173-t009:** Products Purchased by Visitors.

Item	*n*	%
Santiago cake	198	27.97
Cheese	134	18.93
Cakes	82	11.58
Wine	43	6.07
Liquor	41	5.79
Empanada	33	4.66
Bread	29	4.04
Oher typical products	51	7.20
Other products	97	13.70
Total answers	708	100
NS/NR	2081	

Number of answers (*n*) and percentage (%) in relationship with Total answers.

**Table 10 ijerph-17-06173-t010:** Food Products Purchased by Visitors.

Food Choice *n* = 533	Galiza %	Portugal %	Brazil %	Spain %
Santiago Pie	11.3	33.7	31.5	41.8
Cheese	5.7	16.9	20.4	31.5
Wine	0.0	5.6	9.3	12.8
Cakes	32.1	29.2	8.6	16.7
Bread	13.2	6.7	3.7	4.2

Number of answers (*n*) and percentage (%). Differences at level *p* = 0.05.

**Table 11 ijerph-17-06173-t011:** Purchase of Food Products Based on the Number of Times they had Visited the City.

Food Choice	Less One Time/Year *n* = 444	One Time/Year or More *n* = 83
Santiago Pie	40.3	16.9
Cheese	26.6	15.7
Bread	4.5	10.8
Cakes	13.3	24.1

Number of answers (*n*) and percentage (%). Differences at level *p* = 0.05.

**Table 12 ijerph-17-06173-t012:** Places where Visitors Shop.

City District	*n*	%
Old Town	64	26.56
Old Tow-Touristic Shops	64	26.56
Central Market	33	13.69
Supermarkets	72	29.88
Others	8	3.32
Total answers	241	100

Number of answers (*n*) and percentage (%) in relationship with Total answers.

**Table 13 ijerph-17-06173-t013:** Sources of Information on the Commercial Offer Consulted by Visitors.

Sources of Information	*n*	%
Friends, relatives, fellows	15	27.77
People by the street	9	16.67
Official group guide	3	5.56
Hotel staff	7	10.11
Pilgrim/Tourist Office	3	5.56
Internet	8	14.81
Other sources	7	12.96
Total answers	54	100
Don’t know/Do not answer	2027	

Number of answers (*n*) and percentage (%) in relationship with Total answers.

**Table 14 ijerph-17-06173-t014:** Opinions on the Gastronomic Offer (Variety, Quality, Customer Service, and Catering Specifically to Tourists).

Gastronomic Attribute	Positive Consideration %	Negative Consideration %
Variety in offering	19.73	14.58
Suitable price	11.49	6.86
Offering enough	11.84	3.43
Good quality of products	17.32	2.74
Good customer attention	2.74	1.54
Higher touristic emphasis	0	2.74
General valuation	10.63	0.34
Total answer	583 (380 positive and 203 negative)	

Percentage (%) in relationship with Total answers.

**Table 15 ijerph-17-06173-t015:** Proportion of Bars, Restaurants, and Food Stores in Different Areas of the City.

City District	Restaurant *n* (%)	Bar *n* (%)	Food shop *n* (%)	Total *n* (%)
French pilgrim way	28 (8.67)	43 (5.99)	25 (7.79)	96 (7.05)
Portuguese pilgrim way	18 (5.57)	18 (2.51)	14 (4.36)	50 (3.67)
The New District	68 (21.05)	168 (23.40)	61 (19)	297 (21.81)
Old Town	92 (28.48)	180 (25.07)	120 (37.38)	392 (28.71)
Industrial estates (North part of the city)	14 (4.33)	19 (2.65)	8 (2.49)	41 (3.01)
Countryside	22 (6.81)	51 (7.10)	10 (3.12)	83 (6.09)
Remain City	81 (25.08)	239 (33.29)	83 (25.86)	403 (29.59)
TOTAL	323	718	321	1,362

Source: National Classification of Economic Activities [CNAE, for its initials in Spanish] [56]. Number of commercial establishment (*n*) and percentage (%).

**Table 16 ijerph-17-06173-t016:** Top Food Products Identified by Retail and Restaurant Establishments.

Item	%
Drinks, beer, wine	27.54
Menu and take-away meal	24.26
Coffee/tea	9.84
General food	9.18
Meat/butchery	4.26
Fruit/vegetables/grocery	4.26
Octopus/shellfish	3.28
Pizza/pasta	3.28
Pie/cake	2.95
Hamburger, kebab, sandwich	2.62
Bread	2.30
Fish	1.97
Tapas	0.98
Empanada	0.66
Stew/boiled meal	0.66
Without “start food/meal”	1.97
Total (*n* = 305)	100.00

**Table 17 ijerph-17-06173-t017:** Typical Food Consumption Model in Galiza.

Food Items and Their Position (Consumption Preference) from 564 Total Food Items	Consumption (kg/monthly/home). Total Food Consumption(53.4 kg/year/home)	% Consumption from Total Food
1-Fresh Fruit	8.9	16.6
2-Milk	7.1	13.3
6-Fresh Vegetables	4.8	9.1
7-Meat	4.2	7.9
8-Bread	3.6	6.7
9-Fresh Meat	3.3	6.2
18-Potatoes	2.7	5.0
29- Fresh Fish	2.5	2.4
48-Shellfish	0.7	1.3
175-Fresh Octopus	0.1	0.2
316-Freeze Octopus	0.03	0.1

**Table 18 ijerph-17-06173-t018:** Interannual Variation for the Main Foods Consumed in Galiza.

Consumption Coefficient of Variation (%) (Intra-Annual Variation) for Different Food Items from Table 16
10.8—Meat
9.5—Fish
33.8—Shellfish
20.6—Octopus
35.3—Shellfish/Total food
11.4—(Meat/Fish)/Total food

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
