# Peer review of "Gastronomy and Tourism: Socioeconomic and Territorial Implications in Santiago de Compostela-Galiza (NW Spain)"

_ijerph, 2020, doi:10.3390/ijerph17176173_

Round 1
Reviewer 1 Report
The article addresses an interesting and relevant topic for tourism studies. However, there are some aspects of the article that must be improved to be accepted for publication.
Introduction: The theoretical contribution of the study is not clear in this section. The authors concentrated their efforts to explain the context of the study while leaving aside the research gaps. I recommend to explicitly explain these research gaps and the theoretical/methodological contributions of the study supported by recent references (2016-2020).
Literature Review: Even though the authors decided not to include this section in the article, I strongly suggest you to, at least, add the theoretical approach through which you will analyze the phenomenon of interest.
Discussions: This section heavily relies on the context. It is crucial to explain how the results of the study can expand our knowledge regarding gastronomic tourism.
Conclusions: It is important to include the theoretical and managerial implications of the study to the tourism discipline.
References: Please add more references from recent academic articles.
Author Response
First at all, thanks a lot for the comments. Their contribution improve, in a great manner, the quality of manuscript.
Author’s comments
Introduction
Theoretical contribution of the study is now explained in this section. We refer to a broad, wide-range approximation for gastronomic tourism research, more than a specific item research-based, e.g. market segmentation, one food item tourism promotion, …. A literature review (recent academic articles) was done in order to justify the theoretical contribution of our paper. We characterized our research work under theoretical methodological proposal about the need of a holistic and sustainable approximation in gastronomic tourism research (Gilbert, 1990 [25]; Jafari, 2005 [26]; Yeoman and McMahon-Beatte, 2016 [24]; Costa Beber and Gastal, 2020, [27]), linked traditional point of views with new ones. In this section we discussed about different methodological/ conceptual approximations by different authors, and we present our research under the holistic perspective. In this way, our work run about the analysis of food offering, tourist food consumption, spatial distribution of “gastronomic places/locations in the city”, and the study compares this facts with daily-life food consumption and the food potential production (foodshed analysis). Under our knowledge the researching results point out the relevance of a holistic approximation in the gastronomic tourism research, both from methodological point of view (researching approximation), and from the potential implications in the management field for a sustainable tourism development.
Discussion
A new paragraph was added. We are trying to explain why our holistic approximation to gastronomic tourism analysis in Santiago de Compostela could contribution to the understanding, in a better way, the whole implication of tourism fact, and its relationship with sustainability concept: economic, social and environmental potential impacts.
Conclusions
Several paragraphs were added to link the research results and theoretical and managerial implications. In this way, we consider that the conceptual approximation used in our research, holistic one, field research and the manage of several data sources from free- access data-base (governmental agencies), presents a very useful contribution to the field of tourism research. New literature review was done, and its analysis supports the necessity and useful of a holistic/sustainability perspective application for tourism fact research. And the results founded show the main way to do, in present-future, in relationship with the necessity of a more wide-range view for tourism management.
Literature review/References
New 32 references were added, manly from the last 6 years-period.
Reviewer 2 Report
Thank you for inviting me to read this manuscript, which presents an investigation of the contribution that gastronomic tourism brings to regional sustainable development. The theme is interesting and I enjoyed reading the paper.
The authors conclude that gastronomic tourism does not contribute to sustainable development in the region (a very broad focus). My main suggestion for the improvement of this manuscript is that authors scrutinize critically their use of terminology and if/when considered relevant to the research question, the authors must explain clearly, the meaning of these terms in the context of this specific study. By making the effort to explain what various concepts mean, the authors can narrow the focus of their research to a manageable scope. For example, the very first assumption of the study is that gastronomic tourism may contribute to sustainable regional development. However, the explanation of what the authors consider that the term “gastronomic tourism” means in the context of this study, is missing. As a result, the reader is left confused with regards of what is being investigated: gastronomic experiences, chain of suppliers involved, demand for gastronomy, social construction of gastronomic encounters, authenticity of gastronomic experiences, etc. Other terms which need clear explanations in the context to which they refer, which is tourism in Santiago de Compostela: holistic approach, “the (?) sustainability of tourism”, “social, economic and territorial development of the host community”, holistic relationship, gastronomic touristic activity, “normal” consumption of the local population, etc (all from the introduction section). The theoretical perspective used in defining terminology, is also helpful for a more precise positioning of the study, which must be clear in order for the manuscript to become publishable. Claims about contributions of one or another activity to sustainable development, need to be precise in order to be meaningful. Ignoring to bring such explanations, makes the claims vulnerable to criticism.
At the moment, it seems that authors combine various theoretical perspectives, mixing management, marketing and economic perspectives, with cultural approaches of experiential elements. However, there is no explanation of the reasons why a combination of various perspectives can better explain the studied phenomena than a single perspective (for example supply and demand for gastronomy). An ample theoretical argumentation, would help ground the study and would make its conclusions more credible.
Furthermore, it is important that the terms of comparison used for analysis (link to the methodology) are Clearly explained and theoretically argued, together with an explanation of what the authors consider to be a good or a bad contribution to the sustainability of tourism in the region. If the supply and demand for food Would match, would this secure a social, economic and territorial development in the region? Would it bring sustainability in tourism? These are critical questions which authors are encouraged to explore further in their paper.
Without a clear positioning of the research in the theoretical landscape of tourism production and consumption, and a more precise discussion based on clear analytical criteria, it is difficult to see the theoretical or empirical contribution of this manuscript. Therefore, I recommend that the authors take a more consistent critical approach to their use of theoretical concepts, continue to work and review the manuscript and resubmit it at a later stage.
Author Response
First at all, thanks a lot for the comments. Their contribution improve, in a great manner, the quality of manuscript.
Author’s comments
Meaning of different concepts
New references and analysis of them were added in the Introduction section in order to clarify the concepts of sustainable development, gastronomic tourism and its relationships with the aim of our research. Definitions from UNWTO, 2017 [5] or Montecinos, 2016 [6] were used.
In the same way, new paragraphs were added to contextualize the above-mentioned concepts in Santiago de Compostela tourism- fact researching.
Theoretical contribution of the study is now explained in this Introduction section. We refer to a broad, wide-range approximation for gastronomic tourism research, more than a specific item research-based, e.g. market segmentation, one food item tourism promotion, …. A literature review was done to justify the theoretical contribution of our paper. We characterized our research work under theoretical methodological proposal about the need of a holistic and sustainable approximation in gastronomic tourism research (Gilbert, 1990 [25]; Jafari, 2005 [26]; Yeoman and McMahon-Beatte, 2016 [24]; Costa Beber and Gastal, 2020, [27]), linked traditional point of views with more recent ones. In this section we discussed about different methodological/ conceptual approximations by different authors, and we present our research under the holistic perspective framework.
Through the conclusions section new paragraphs were added in order to argue the relevance of the obtained results in relationship with the holistic methodological analysis done, and with the gastronomic tourism landscape founded. We discuss the implications of the results of our research in relationship with “Tourism and the Sustainable Development Goals – Journey to 2030” (UNWTO 2017)[5], in order to check the contribution of Santiago de Compostela tourism-fact to the five goals describe in UNWTO (2017) highlights.
Round 2
Reviewer 1 Report
I agree with the changes made by the authors.